# Deodorant Activity of Black Cumin Seed Essential Oil against Garlic Organosulfur Compound

**DOI:** 10.3390/biom11121874

**Published:** 2021-12-14

**Authors:** Yuri Yoshioka, Shinichi Matsumura, Masanori Morimoto

**Affiliations:** 1Natural Products, Inabata Koryo Co., Ltd., Yodogawa, Osaka 532-0027, Japan; y-yoshioka@inabatakoryo.co.jp (Y.Y.); shin-matsumura@inabatakoryo.co.jp (S.M.); 2Department of Applied Biological Chemistry, School of Agriculture, Kindai University, Nara 631-8505, Japan

**Keywords:** black cumin seed essential oil, deodorant activity, garlic organosulfur compounds, *Nigella sativa* L.

## Abstract

The deodorant activity of black cumin (*Nigella sativa* L.) seed, a spice used to flavor curry and vegetable foods in Southwest Asia, against garlic (*Allium sativum* L.) organosulfur compounds related to human malodor was evaluated. Black cumin seed essential oil showed remarkable deodorant activity against garlic essential oil. The mode of action of this deodorant activity was presumed to be that black cumin seed essential oil covalently reacted with the organosulfur compounds in garlic. Therefore, thymoquinone, which is a major constituent in black cumin seed essential oil, and allyl mercaptan, which is one of the organosulfur compounds produced by cutting garlic, were reacted in vitro, and the products were purified and elucidated using spectroscopic data. As a result, these substances were identified as different allyl mercaptan adducts to dihydrothymoquinone. This chemical reaction was presumed to play a key role in the deodorant activity of black cumin seed essential oil.

## 1. Introduction

Black cumin (*Nigella sativa* L.) seed is a species of the Ranunculaceae family, and thymoquinone is a major component of black cumin seed essential oil. Black cumin seed essential oil (BCO) exhibits various beneficial effects, including restoring the antioxidant defense system, increasing the activity of antioxidant-related enzymes, decreasing inflammatory biomarkers, suppressing pro-inflammatory mediator production, preventing endothelial dysfunction, and improving hepatic, kidney, heart, and immune system functions [1].

Garlic (*Allium sativum* L.) is a species of the Amaryllidaceae family and is used as a food and spice worldwide. This plant exhibits various functions, such as antibacterial, antiviral, antioxidant, and anti-inflammatory effects [2]. However, garlic food products contain various organosulfur compounds that lead to bad breath, and the odor lasts for a long time after ingestion. For this reason, many people often hesitate to eat garlic products [3].

Halitosis and a peculiar garlic odor are common after the ingestion of garlic products. In the oral cavity, garlic fragments result in methyl mercaptan (MM) and hydrogen sulfide, leading to bad breath from these food products. MM and hydrogen sulfide are common causes of malodorous breath exhaled from the normal human body. In addition, other organosulfur compounds, including allyl mercaptan (AM) and allyl methyl sulfide (AMS), result from digestion in the stomach, and these chemicals are released as exhaled breath and excreted through sweat glands and sebaceous glands by the bloodstream, leading to body odor [4].

Since MM is a primary factor for halitosis, it has been studied extensively [1]. There have been several reports concerning halitosis prevention using plant materials against MM. Thyme (*Thymus vulgaris* L.) and rosemary (*Rosmarinus officinalis* L.) have both shown remarkable deodorant activity against MM [5]. Phenolic compounds, such as chlorogenic acid, caffeic acid, and tea catechin, can also be used as deodorizing agents against MM [6,7]. In addition, BCO, which was used in this study, has been reported to show strong deodorant activity against MM. The active ingredient of BCO is thymoquinone, whose deodorant activity exceeds that of tea catechin [8]. The deodorizing mechanism of BCO is presumed to be related to the binding of the mercapto group of MM [9].

After garlic ingestion, human exhalation acquires a peculiar odor. Furthermore, garlic is absorbed into the body by digestion, causing a bad smell to be released from the whole body, including exhaled breath. Previous studies on exhaled breath components related to garlic ingestion have revealed several volatile organosulfur compounds associated with the characteristic odor of garlic in exhaled breath, such as diallyl disulfide (DADS) and AMS [10,11]. DADS and AMS are detected in high concentrations in human exhaled breath immediately after garlic ingestion [11,12,13]. The digestion of AMS takes several hours, and AMS has been reported to be responsible for the persistent odor of garlic in exhaled breath.^14)^ In addition, AMS and DADS have been reported as body odors on the skin surface after the ingestion of garlic [14]. However, there are few reports of deodorant activity against garlic odor, and this mechanism has not been clarified.

Common halitosis is generated from food fragments in the oral cavity, and there have been many reports on deodorant activity in mimicked oral environments, in which the pH is neutral and food is present for a short time. However, when it comes to halitosis after garlic ingestion, it is necessary to assume the environmental conditions of the stomach, that is, where the pH is acidic and there is residual time regarding the presence of food. There have been several reports on gastric emptying time. Degen et al. compared men and women and reported that the median gastric emptying time (T 1/2) was approximately 150 min for men and approximately 200 min for women [15]. Hellmig et al. reported that the median gastric emptying time (T 1/2) was 80.5 ± 22.2 min for liquids and 127 min for solids, although not normally distributed [16].

Accordingly, we evaluated the deodorant activity of both conditions, with an oral cavity mimic at pH 7.0 and a gastric environment mimic at pH 1.2. The duration of the deodorant test was decided in consideration of the residual time of food in the stomach, as per a previous paper. We evaluated whether BCO, which has a deodorizing effect against MM, is also effective for the volatile organosulfur compounds of garlic that cause bad breath after garlic ingestion. Furthermore, the reaction products between AM and thymoquinone, which is a major constituent in BCO, were isolated and the chemical structures were elucidated to help us better understand the deodorizing mechanism.

## 2. Materials and Methods

### 2.1. General Experimental Procedures

One- and two-dimensional nuclear magnetic resonance (NMR) spectra were obtained using an Avance 400 instrument (Bruker Co., Ltd., Bremen, Germany), with a solvent signal of CDCl_3_ as the internal reference. Gas chromatography-mass spectrometry (GC-MS) analyses were performed using an Agilent 7890A-5975C MSD instrument (Agilent Technologies, Santa Clara, CA, USA). Desorption electron ionization (DEI-MS) was measured using JEOL JMS-K9 (JEOL Co. Ltd., Tokyo, Japan). Headspace-gas chromatography (HS-GC) analyses were performed using an Agilent 6890 GC instrument (Agilent Technologies), which was equipped with a pulsed flame photometric detector (PFPD, model 5380, Agilent Technologies). Liquid chromatography-mass spectrometry (LC-MS) analyses were performed using an ultra-performance liquid chromatography-mass spectrometry system (LCMS-8050, Shimadzu, Kyoto, Japan), which was equipped with Lab Solutions Version 5.75. Flash column chromatography was performed on an Isolera One instrument (Biotage, Uppsala, Sweden).

### 2.2. Botanical Essential Oils

Black cumin (*N. sativa*) seed essential oil (BCO, thymoquinone content of 41.5% (*v*/*v*)) was purchased from Arjuna Natural (Cochin, India). Garlic (*A. sativum*) essential oil (GEO) was purchased from Plant Lipids (Cochin, India). Both BCO and GEO were manufactured industrially by steam distillation.

### 2.3. Chemicals

Allyl mercaptan (AM, purity 84.6%), allyl methyl sulfide (AMS, purity 99.1%), diallyl sulfide (DAS, purity 99.7%), allyl methyl disulfide (AMDS, purity 95.6%), dimethyl trisulfide (DMTS, purity 99.6%), and diallyl disulfide (DADS, purity 84.1%) were purchased from Tokyo Chemical Industry Co., Ltd. (Tokyo, Japan). Allyl methyl trisulfide (AMTS) and diallyl trisulfide (DATS) were isolated from GEO.

### 2.4. Isolation of AMTS and DATS from GEO

GEO (100 g, including AMTS 10.0% and DATS 45.2%) was distilled under reduced pressure (temperature, 70 °C; vacuum, 0.5 kPa for 1 h), and garlic distillation oil (GDO, 24.1 g, including AMTS 15.7% and DATS 9.4%) and its residue of GEO after distillation (RGO, 68.4 g, including AMTS 9.4% and DATS 67.5%) were obtained. GDO (8 g) was then loaded into a preparative high-performance liquid chromatography (HPLC) system (Shimadzu), using the following parameters: column, L-column ODS (separation column 20 mm I.D. × 250 mm, particle size 5 µm, guard column 10 mm I.D. × 20 mm, particle size 5 µm, Chemicals Evaluation and Research Institute, Tokyo, Japan); column temperature, ambient; mobile phase, water:acetonitrile = 25:75; flow rate, 18.9 mL/min; injection volume, 400 µL; and a photodiode array detector. UV absorption was monitored at 190 and 214 nm. Under these conditions, the two fractions with peaks at 8 min and 10 min were collected and concentrated to obtain GDO Fraction 1 (yield 3.24 g) and GDO Fraction 2 (yield 0.52 g), respectively. GDO Fraction 1 was again purified in the same preparative HPLC system with a different method using a mobile phase of water:acetonitrile = 30:70 and an injection volume of 50 µL. The fraction with a peak at 10.5 min was collected and concentrated to obtain GDO Fraction 3 (yield 0.33 g) [17,18,19]. GDO Fraction 2 and GDO Fraction 3 were identified as DATS (purity 95.8%) and AMTS (purity 98.4%) by GC-MS analysis, respectively. Furthermore, these isolated compounds were identified using NMR spectra.

Allyl methyl trisulfide (AMTS): ^1^H NMR (400 MHz) CDCl_3_ δ (ppm): 5.89 (1H, ddt, *J* = 16.9, 9.97, 7.34 Hz), 5.23 (1H, dd, *J* = 16.9, 1.38 Hz), 5.20 (1H, dd, *J* = 9.97, 1.38 Hz), 3.51 (2H, d, *J* = 7.34 Hz), 2.56 (3H, s). ^13^C NMR (100 MHz) CDCl_3_ δ (ppm): 132.7, 119.1, 41.5, 22.8. EIMS (70 eV) *m*/*z* (relative intensity, %); 152 (M^+^, 10.7), 111 (17.0), 87 (100), 79 (15.1), 73 (82.1), 64 (12.3), 45 (43.5).

Diallyl trisulfide (DATS): ^1^H NMR (400 MHz) CDCl_3_ δ (ppm): 5.88 (2H, ddt, *J* = 16.8, 9.79, 7.28 Hz), 5.23 (2H, dd, *J* = 16.8, 1.38 Hz), 5.20 (2H, dd, *J* = 9.79, 1.38 Hz), 3.51 (4H, d, *J* = 7.28 Hz). ^13^C NMR (100 MHz) CDCl_3_ δ (ppm): 132.7, 119.1, 41.6. EIMS (70 eV) *m*/*z* (relative intensity, %); 178 (M^+^, 8.0), 113 (100), 73 (97.9), 41 (60.1).

### 2.5. GC-MS Analysis

The following GC-MS analytical conditions were used: column, InertCap Pure-WAX (60 m × 0.25 mm I.D., 0.25 mm film thickness, GL Sciences, Inc., Tokyo, Japan); injection temperature, 250 °C; injection volume, 1 µL; split ratio, 80:1; carrier gas flow, 1.2 mL/min (helium, constant flow); temperature program, 50 °C (2 min hold), 50–240 °C at 2.5 °C/min and 240 °C (50 min hold); ion source temperature, 230 °C; and quadrupole temperature, 150 °C. Ionization was performed using the electron impact ionization method at 70 eV (scan range 25–350 *m*/*z*). The constituents of the essential oil were identified using the NIST MS library (NIST 08) (National Institute of Standards and Technology, Standard Reference Data, Gaithersburg, MD, USA) and the Wiley MS Library (Version 7) (Cerno Bioscience, Las Vegas, NV, USA) along with retention indexes. The content of each component was shown as the area percentage relative to the total area of detected peaks in the GC-MS analysis.

### 2.6. HS-GC Analysis

The following HS-GC analytical conditions were used: column, InertCap Pure-WAX (60 m × 0.25 mm I.D., 0.25 mm film thickness, GL Sciences, Inc.); injection temperature, 250 °C; injection volume, 1 mL (headspace gas); split ratio, 10:1; carrier gas flow, 1.2 mL/min (helium, constant flow); and temperature program, 50 °C (2 min hold), 50–200 °C at 7.5 °C/min and 200–240 °C at 20 °C/min. The constituents of the essential oil were identified based on the retention times of commercial reagents [20].

### 2.7. Evaluation of Deodorant Activity

The deodorant activity of BCO against GEO and the organosulfur compounds in garlic was evaluated using the headspace method. The disintegration test solution (pH 1.2) used was diluted, 1st fluid for disintegration test, pH 1.2/1st fluid for dissolution test, pH 1.2 (×10) (Wako 066-06441, Fujifilm Wako Pure Chemical, Tokyo, Japan), with ion-exchanged water (pH 1.2 test solution). The phosphate buffer (pH 7.0) used was an instant phosphate buffer (RM-102-3 L, LSI Medience Corporation, Tokyo, Japan). GEO and the eight major organosulfur compounds in garlic were prepared with ethanol to defined concentrations (GEO, 300 ng/µL; AM, AMS, DAS, AMDS, DMTS, and DADS, 10 ng/µL; AMTS, 500 ng/µL; and DATS, 1000 ng/µL). These concentrations of organosulfur compounds were adjusted under PFPD saturation. The BCO ethanolic solution (1 mL) and pH 1.2 test solution or pH 7.0 buffer (5 mL) were mixed in a 30 mL glass vial and sealed with a silicone cap. After adding 50 µL of GEO or organosulfur compound ethanolic solution, the glass vial was shaken strongly for 1 min. The mixture was then left standing at 37 °C for 60, 120, and 180 min with GEO; 10 min with AM; 120 min with AMS, DAS, and AMDS; and 180 min with DMTS, DADS, AMTS, and DATS. The organosulfur compounds released to the headspace of the glass vial were collected with a gastight syringe, and the sampled headspace gas (1 mL) was injected into a GC equipped with a PFPD to calculate the peak area of the target organosulfur component. The deodorant activities of the test chemicals were calculated using the following equation [21]:(1)$$\mathrm{Deodorant}\;\mathrm{activity}\;\left(\%\right)=\frac{\left(C-S\right)}{C}\times 100\;(C,\;\mathrm{peak}\;\mathrm{area}\;\mathrm{of}\;\mathrm{control};\;S,\;\mathrm{peak}\;\mathrm{area}\;\mathrm{of}\;\mathrm{sample}).$$

### 2.8. Isolation and Identification of Reactants from BCO and AM

BCO (0.5 g) was dissolved with ethanol (20 mL) and AM (500 µL), mixed, and then kept at 55 °C for 45 min. Then, the ethanol was distilled away under reduced pressure at 50 °C using a rotary evaporator (yield 0.98 g). The yellowish oil was fractionated by silica gel flash chromatography to obtain the subfraction (yield 0.62 g). The separating conditions were as follows: column, SNAP KP-Sil Cartridge 25 g (normal silica gel column, particle size 50 µm, Biotage, Osaka, Japan); mobile phase, hexane-ethyl acetate (gradient, 5–25% *v*/*v*, ethyl acetate in hexane); and detection at UV 254 nm. The obtained subfraction (82 mg) was purified by silica gel flash chromatography with hexane-ethyl acetate (gradient, 5–22% *v*/*v*, ethyl acetate in hexane) to obtain compound (**1**) (3 mg), compound (**2**) (8 mg), and compound (**3**) (5 mg).

Compound (**1**): yellowish oil; ^1^H NMR (400 MHz) CDCl_3_ δ (ppm): 6.90 (1H, s), 6.54 (1H, s), 5.82 (1H, ddt, *J* = 18.1, 10.0, 7.2 Hz), 5.01 (1H, d, *J* = 10.0 Hz), 4.96 (1H, dd, *J* = 18.1, 1.3 Hz), 4.31 (1H, br. s), 3.83 (1H, sext, *J* = 7.0 Hz), 3.23 (2H, d, *J* = 7.2 Hz), 2.19 (3H, s), 1.38 (6H, d, *J* = 7.0 Hz). ^13^C NMR (100 MHz) CDCl_3_ δ (ppm): 149.8, 147.2, 135.0, 133.0, 121.6, 120.8, 118.2, 117.5, 39.6, 22.4, 20.9 (dup.), 16.3; DEIMS (70 eV) *m*/*z* (rel. int. %) 238 (M^+^, 15.4), 195 (100), 180 (70.3), 162 (19.5), 155 (16.7), 57 (9.6).

Compound (**2**): yellowish oil; ^1^H NMR (400 MHz) CDCl_3_ δ (ppm): 6.81 (1H, s), 6.70 (1H, s), 5.80 (1H, ddt, *J* = 16.8, 9.9, 7.2 Hz), 4.97 (1H, d, *J* = 9.9 Hz), 4.91 (1H, dd, *J* = 16.8, 1.3 Hz), 4.45 (1H, s), 3.24 (1H, sext, *J* = 7.0 Hz), 3.22 (2H, d, *J* = 7.2 Hz), 2.40 (3H, s), 1.20 (6H, d, *J* = 7.0 Hz). ^13^C NMR (100 MHz) CDCl_3_ δ (ppm): 148.9, 146.7, 133.0, 132.3, 125.2, 118.2, 118.1, 114.9, 38.6, 27.6, 22.4 (dup.), 14.1; DEIMS (70 eV) *m/z* (rel. int. %) 238 (M^+^, 73.8), 203 (18.5), 197 (100), 164 (62.8), 163 (48.3), 99 (24.4), 91 (27.5), 67 (30.3).

Compound (**3**): yellowish oil; ^1^H NMR (400 MHz) CDCl_3_ δ (ppm): 6.83 (1H, s), 6.70 (1H, s), 5.74 (1H, ddt, *J* = 13.2, 6.4, 7.2 Hz), 5.07 (1H, d, *J* = 6.4 Hz), 5.03 (1H, dd, *J* = 13.2, 1.4 Hz), 4.39 (1H, s), 3.25 (1H, sext, *J* = 7.1 Hz), 3.07 (2H, d, *J* = 7.2 Hz), 2.70 (2H, t, *J* = 7.2 Hz), 2.52 (2H, t, *J* = 7.1 Hz), 2.41 (3H, s), 1.78 (2H, q, *J* = 7.3 Hz), 1.20 (6H, d, *J* = 6.8 Hz). ^13^C NMR (100 MHz) CDCl_3_ δ (ppm): 148.8, 146.9, 134.2, 132.3, 125.0, 118.8, 117.1, 114.8, 34.6, 34.3, 29.5, 28.9, 27.6, 22.4 (dup.), 14.1; DEIMS (70 eV) *m/z* (rel. int. %) 312 (M^+^, 14.4), 271 (15.9), 269 (100), 87 (16.0), 73 (53.0), 67 (28.2).

### 2.9. Reaction Monitoring between Thymoquinone and AM

The fixed amount of ethanolic solution (1 mL) of thymoquinone (400 μg) and the fixed amount of ethanolic solution of AM (0.05 mL) were added to the buffer solution (pH 1.2 or 7.0, 5 mL), sealed, and kept at 37 °C for 10, 30, or 60 min. The buffer solution was similar to that used in the evaluation of the deodorant activity. The amount of thymoquinone (400 μg) was equivalent to 1 mg of BCO. In this test, the applied concentrations of AM were 50, 5, and 0.5 µg/mL.

The following analysis conditions were used: column, L-column 2 ODS (2.1 mm I.D. × 150 mm, particle size 2.0 μm); detector, UV, 254 nm; flow rate, 0.4 mL/min; mobile phase, water:acetonitrile = 27:73; oven temperature, 40 °C; and injection volume, 2 μL. The electrospray ionization mass spectrometry (ESI-MS) parameters were as follows: interface temperature, 300 °C; desolvation line (DL) temperature, 250 °C; heat block temperature, 400 °C; nebulizer gas flow rate, 3 L/min; heating gas flow rate, 10 L/min; and drying gas flow rate, 10 L/min. The multiple reaction monitoring (MRM) mode was used to monitor the transition of compound (**1**) (*m*/*z*: 237.10 > 196.10), compound (**2**) (*m*/*z*: 237.10 > 196.10), and compound (**3**) (*m*/*z*: 310.10 > 195.05). The retention times of compounds **1**–**3** were 2.33, 2.21, and 3.24 min, respectively.

### 2.10. Statistical Analysis

All data were analyzed statistically by one-way analysis of variance followed by multiple comparison tests using the Tukey–Kramer test. The analyses were performed using Bell Curve for Excel (Version 3.21, Social Survey Research Information Co., Ltd., Tokyo, Japan) an add-in software for Excel.

## 3. Results and Discussion

### 3.1. Organosulfur Compound Composition of GEO

The major organosulfur compounds in GEO according to the GC-MS analysis were allyl mercaptan (AM), allyl methyl sulfide (AMS), diallyl sulfide (DAS), allyl methyl disulfide (AMDS), dimethyl trisulfide (DMTS), diallyl disulfide (DADS), allyl methyl trisulfide (AMTS), and diallyl trisulfide (DATS). The area percentages of each organosulfur compound toward the total area of the GC peaks in GEO were 0.14%, 2.00%, 6.24%, 3.90%, 7.17%, 13.2%, 16.6%, and 40.2%, respectively. The major constituent of GEO was DATS (Figure 1). The organosulfur compound compositions of GEO from air-dried and freeze-dried garlic have been found to be characterized by an amount of organosulfur compound (84.3–98.9%) consisting of DATS (37.3–45.9%), DADS (17.5–35.6%), and AMTS (7.7–10.4%) [22]. In essential oil analyses of garlic varieties, the chemical composition has been found to consist of AMS (4.4–12.0%), DAS (1.3–2.4%), DMTS (0.5–2.2%), DADS (27.1–46.8%), AMTS (8.3–18.2%), and DATS (19.9–34.7%) [23]. Based on these reports, the GEO used in this study was considered to be a typical GEO.

### 3.2. Deodorant Activity of Black Cumin Seed Essential Oil (BCO) against GEO

In a 30 mL glass vial, BCO (10 mg) showed remarkable deodorant activity at pH 1.2 at 180 min after treatment. The deodorant activities of AMS, DAS, AMDS, DMTS, DADS, and AMTS were 56.3%, 72.4%, 59.0%, 69.5%, 28.4%, and 32.0%, respectively (Figure 2). The deodorant activities of AMS, DAS, AMDS, and DMTS were time-dependent, but DADS and AMTS were time-independent between 60 min and 120 min (Figure 3). We presume that DADS and AMTS have bulky structures and highly lipophilic properties compared to other compounds, and DADS and AMTS were difficult to volatilize into the headspace (Figure 1).

Unfortunately, AM and DATS were not detectable under the selected analytical conditions. Presumably, the amount of AM in GEO was very low and DATS had a high boiling point, and they could not sufficiently vaporize into the headspace to be detected by GC with a PFPD (Figure 1).

### 3.3. Deodorant Activity of BCO against Organosulfur Compounds

In a 30 mL glass vial, the 0.1 mg treatment of BCO showed remarkable deodorant activity against 0.5 µg of AM (75.7% at pH 7.0, 18.7% at pH 1.2). Under the same conditions, the 10 mg treatment of BCO showed deodorant activity against 0.5 µg of AM (100% at pH 7.0, 70.2% at pH 1.2) (Table 1). The deodorant activity of BCO was dose-dependent, but it was decreased at pH 1.2 compared to pH 7.0. Additionally, the deodorant activity of BCO was effective against all test organosulfur compounds at pH 1.2 (Table 2). This result suggests that BCO may decrease the amount of organosulfur compound derived from garlic in the stomach, and thus reduce halitosis due to belching. As previously mentioned, common halitosis involves oral odors, and many reports have involved tests performed at pH 7.0. However, in this study, we focused on deodorant activity in the stomach rather than the oral cavity. Specifically, we evaluated deodorant activity in a gastric environment at pH 1.2, since the garlic halitosis almost always arises from the decomposition of garlic organosulfur compounds in the stomach. As shown in Table 1, our comparative studies of pH 1.2 and pH 7.0 showed that the deodorant activity against AM tended to be lower under the pH 1.2 condition. Although GEO has low AM content, a considerable amount of AM is detected in the breath of humans who consume garlic [24]. The odor of garlic is known to derive from alliin. When raw garlic is crushed or minced, alliin is converted to allicin by alliinase. However, allicin is unstable and easy to decompose into various organosulfur compounds, such as DADS, DATS, and ajoene. Furthermore, DADS and DATS are converted to AM, followed by conversion to AMS [14,25]. These results suggest that BCO may act as a deodorizing agent, thereby preventing halitosis after garlic meal ingestion.

### 3.4. Isolation and Identification of Reactants between Thymoquinone and AM

A few products of the reaction between thymoquinone and AM were isolated using silica gel flash chromatography with hexane and ethyl acetate as the mobile phase. The elucidation of the structures of these products was performed using ^1^H NMR, ^13^C NMR, and 2D NMR (HSQC, HMBC) spectral data. Three reaction products, compounds (**1**–**3**) were identified in this study. Compounds **1** and **2** were AM added to dihydrothymoquinone, a reduced form of the major natural monoterpene thymoquinone in BCO (Figure 4). Compound **3** had a dithio-moiety from one extra AM added to **2** (Figure 4). This result suggests that the deodorizing mechanism of BCO against AM may have occurred due to the conversion of AM to these chemicals.

A previous report concerning the prevention of bad breath due to garlic used various botanical sources containing polyphenols, such as apple, tea, lemon, parsley, and mint [26]. In our present study, AM was trapped with thymoquinone covalently and converted to a non-volatile phenol compound. However, a similar reaction could not be observed in the other tested sulfides in this study (data not shown). This fact suggested that a thiol moiety was essential to this reaction. A previous report described that some organosulfur compounds are converted to AM [27]; therefore, decreasing the AM amount may prevent AMS generation in the body.

A comparison of the effects of non-fat milk and whole milk on the deodorant activity of garlic reported that whole milk was more effective. However, whole milk showed the highest efficacy in comparison with water, whole milk, and 10% casein sodium aq., and it was presumed that the fat contained in whole milk suppressed the volatility of the garlic-derived DAS and AMS [13]. Similarly, the deodorant activity of BCO presumed that not only covalent trapping between thymoquinone and AM but also high hydrophobicity interactions occurred in this study.

### 3.5. Reaction Monitoring between Thymoquinone and AM

The time-dependent reactions between thymoquinone and AM at pH 7.0 were measured. Although the amounts of thymoquinone and AM did not decrease drastically, the amounts of their adducts, compounds **2** and **3,** increased in a time-dependent manner (Figure 5a–c). Similarly, the amounts of AM and thymoquinone did not decrease drastically, but the amounts of their adducts, compounds **1**–**3,** increased in a time-dependent manner at pH 1.2 (Figure 5d–f). The amounts of reactants **1**–**3** were dependent on the acidic conditions in this study (Table 3).

When 50 μg of AM was added to 400 µg of thymoquinone at pH 1.2, compound **1** occurred in concentrations of 4.05, 8.12, and 10.08 ng/mL at 10, 30, and 60 min after mixture, respectively. At pH 7.0, the concentrations of compound **1** were all not detected (N.D.) at 10, 30, and 60 min after mixture, respectively. Similarly, the concentrations of compound **2** were 1.81, 2.96, and 5.66 ng/mL at pH 1.2 at 10, 30, and 60 min, respectively. At pH 7.0, the concentrations of compound **2** were 422.69, 605.32, and 732.26 ng/mL at 10, 30, and 60 min, respectively. The concentrations of compound **3** were 0.08, 0.22, and 0.68 ng/mL at pH 1.2 and 163.23, 231.97, and 261.43 ng/mL at pH 7.0 at 10, 30, and 60 min, respectively.

On the other hand, when 5 μg of AM was added, at pH 1.2, compound **1** occurred in concentrations of 0.14, 0.16, and 0.28 ng/mL at 10, 30, and 60 min after mixture, respectively. At pH 7.0, the concentrations of compound **1** were all N.D. at 10, 30, and 60 min after mixing, respectively. Similarly, the concentrations of compound **2** were 0.02, 0.06, and 0.14 ng/mL at pH 1.2 at 10, 30, and 60 min, respectively. At pH 7.0, the concentrations of compound **2** were 5.58, 18.42, and 28.38 ng/mL at 10, 30, and 60 min, respectively. The concentrations of compound **3** were N.D., N.D., and 0.02 ng/mL at pH 1.2 and 4.31, 10.27, and 11.29 ng/mL at pH 7.0 at 10, 30, and 60 min, respectively. When 0.5 μg of AM was added, compound **1** was N.D. at both pH 1.2 and pH 7.0. Similarly, compound **2** was N.D. at pH 1.2 at 10, 30, and 60 min, respectively. At pH 7.0, the concentrations of compound **2** were 0.26, 1.42, and 2.13 ng/mL at 10, 30, and 60 min, respectively. The concentrations of compound **3** were all N.D. at pH 1.2 and 0.21, 0.50, and 0.70 ng/mL at pH 7.0 at 10, 30, and 60 min, respectively.

The reaction between AM and thymoquinone was promoted at pH 7.0 in comparison with pH 1.2. These results were similar to those of the deodorant evaluations using HS-GC.

## 4. Conclusions

The results obtained for the deodorant activity of BCO against garlic odor suggest that this botanical resource applies to the control of oral malodor after consuming garlic meals. Previous papers about deodorant activity had evaluated this under oral environmental conditions of pH 7.0, while this study also evaluated gastric environmental conditions of pH 1.2. In this study, we discovered that BCO was effective in suppressing bad breath after intaking a garlic meal. We presumed that a part of the mechanism that is a major constituent of BCO, thymoquinone, trapped AM, which is an organosulfur produced from garlic meals in the human body causative of garlic bad breath. However, this deodorant activity of BCO did not overwhelm the odor, and the other non-reacted garlic organosulfur compounds remained in the human body. These residual intact garlic organosulfur compounds, such as DADS and AMTS, may have various valuable bioavailability in the human body. Recently, we showed that garlic organosulfur compounds had the potential to slow the procession of Alzheimer’s disease by the inhibition of related enzymes located in the brain [28]. Although the deodorant activity of BCO is effective in controlling oral malodor from garlic meals, it is better to adjust the intake time to interval eating for garlic bioavailability. It is a critical point from a good human health point of view that the bioavailability of garlic organosulfur compounds increases and the deodorant activity of BCO follows. Finally, research on the deodorizing mechanisms of BCO is likely to continue in the future. People who consume garlic products for health benefits, such as revitalizing cells and the bloodstream, could prevent bad breath due to the odor of garlic by taking BCO.

## Figures and Tables

**Figure 1 biomolecules-11-01874-f001:**
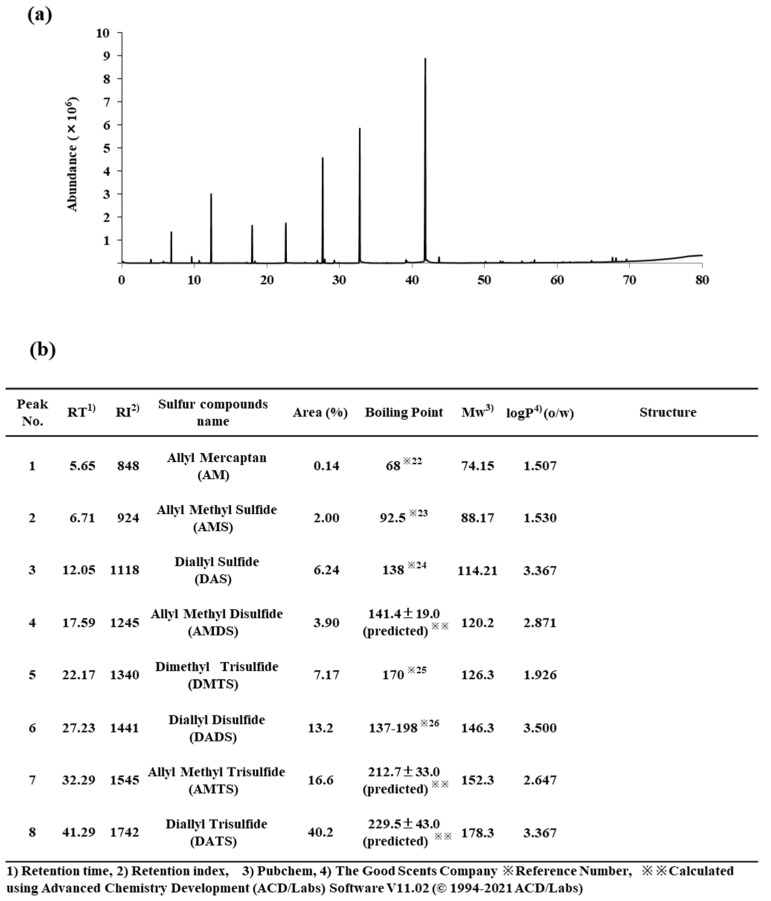
Organosulfur compound composition of GEO: (**a**) GC-MS total ion chromatogram, (**b**) contents and structures of organosulfur compounds in GEO.

**Figure 2 biomolecules-11-01874-f002:**
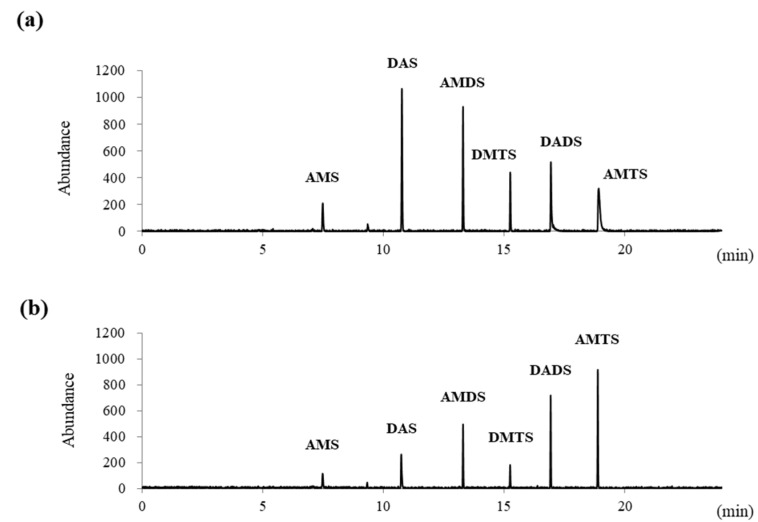
PFPD chart of deodorant activity of BCO against 15 µg of GEO at pH 1.2: (**a**) without BCO, (**b**) 10 mg of BCO added to GEO for 180 min.

**Figure 3 biomolecules-11-01874-f003:**
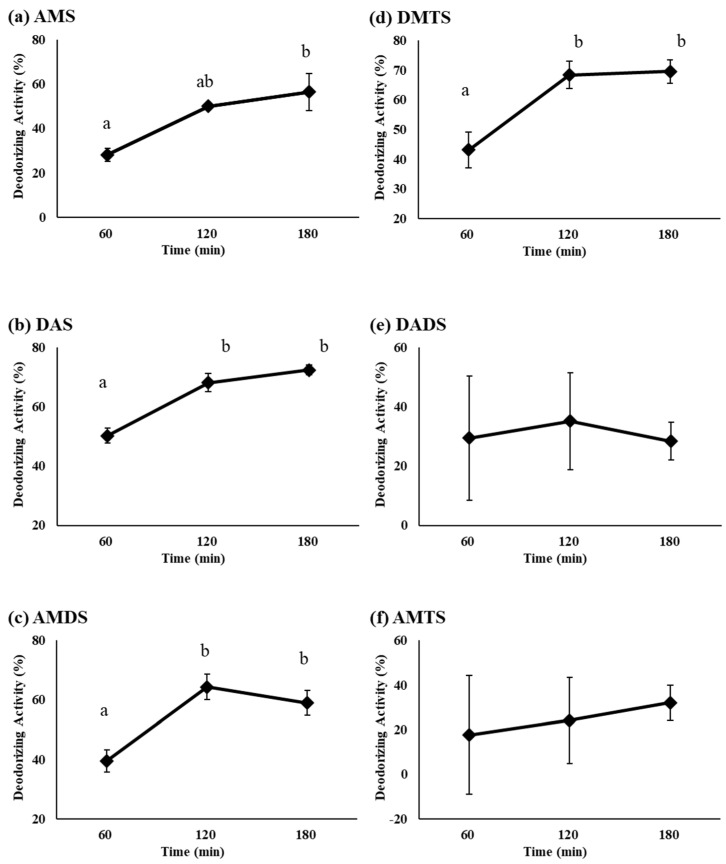
Deodorant activity of BCO against organosulfur compounds in GEO at pH 1.2: (**a**) AMS, (**b**) DAS, (**c**) AMDS, (**d**) DMTS, (**e**) DADS and (**f**) AMTS. Dosage of BCO, 10 mg, dosage of GEO, 15 µg. Values were shown as means with standard error as error bars (*n* = 3) with different letters to show significant differences (*p* < 0.05).

**Figure 4 biomolecules-11-01874-f004:**
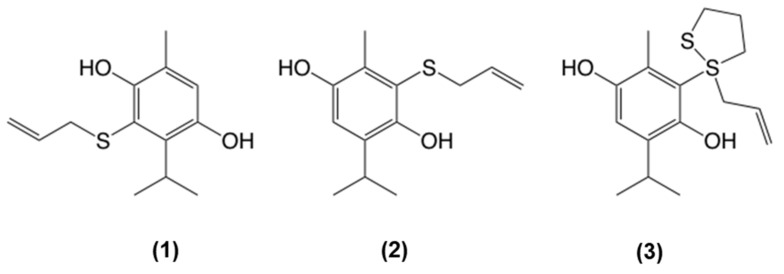
The structure of reactants between BCO and AM. **1**: 3-Allylthiodihydrothymoquinone, **2**: 2-Allylthiodihydrothymoquinone, **3**: 2-(1-Allyl-1,2-dithiolanyl)-dihydrothymoquinone.

**Figure 5 biomolecules-11-01874-f005:**
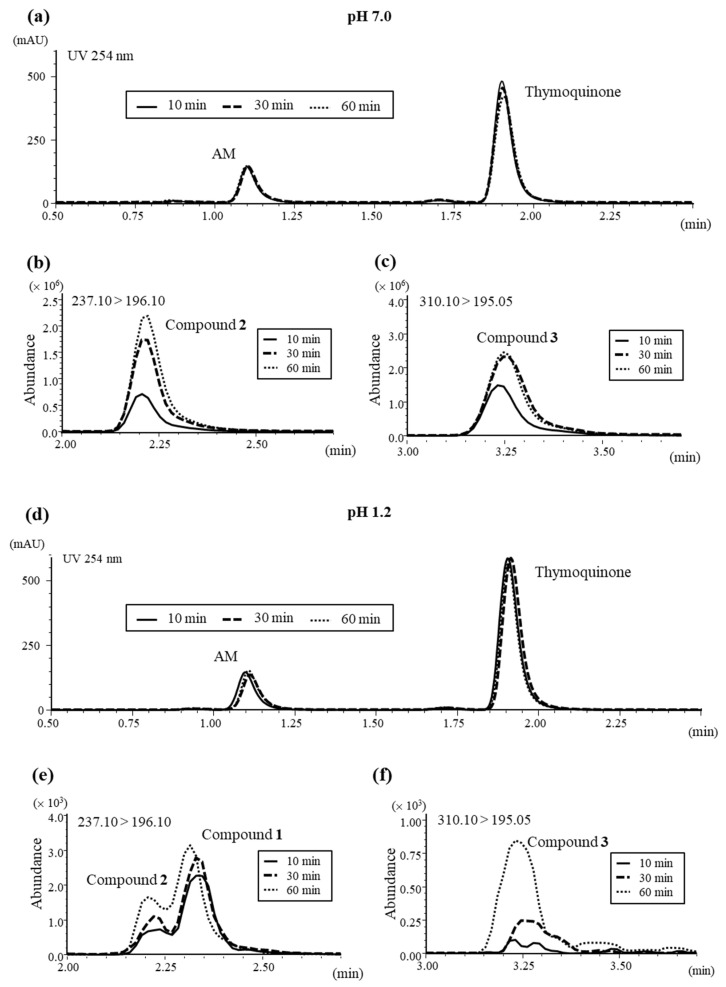
The chromatograph after reaction with 400 µg of thymoquinone and AM: (**a**) HPLC chart at 254 nm of thymoquinone and AM after reaction at pH 7.0, (**b**) MS chromatograph (MRM 237.10 > 196.10) after reaction at pH 7.0, (**c**) MS chromatograph (MRM 310.10 > 195.05) after reaction at pH 7.0, (**d**) HPLC chart at 254 nm of thymoquinone and AM after reaction at pH 1.2, (**e**) MS chromatograph (MRM 237.10 > 196.10) after reaction at pH 1.2, (**f**) MS chromatograph (MRM 310.10 > 195.05) after reaction at pH 1.2. All experiments were performed at 37 °C.

**Table 1 biomolecules-11-01874-t001:** Deodorant activity of black cumin seed oil (BCO) against 0.5 µg of allyl mercaptan (AM).

Assay Condition	Dosage of BCO (mg)	Deodorant Activity ± SD (%)
pH 7.0	0.1	75.7 ± 9.30
	10	100 ± 0.00
pH 1.2	0.1	18.7 ± 16.7
	10	70.2 ± 13.7

Incubation time; 10 min. Each value represents the mean ± SD (*n* = 3).

**Table 2 biomolecules-11-01874-t002:** Deodorant activity of each dosage of black cumin seed oil (BCO) against sulfur compounds at pH 1.2 assay buffer.

Compounds	Deodorant Activity ± SD (%)		
	5 mg of BCO	10 mg of BCO	20 mg of BCO
AMS	31.6 ± 2.72	32.8 ± 11.4	49.5 ± 8.14
DAS	58.9 ± 1.27	57.0 ± 4.58	81.2 ± 2.11
AMDS	22.7 ± 7.38	65.1 ± 6.76	87.2 ± 2.8
DMTS	61.8 ± 11.0	68.8 ± 5.00	85.9 ± 3.23
DADS	43.6 ± 6.52	45.3 ± 11.0	100 ± 0.00
AMTS	58.2 ± 4.42	82.5 ± 2.21	100 ± 0.00
DATS	49.4 ± 2.81	73.7 ± 1.59	100 ± 0.00

Each value represents the mean ± SD (*n* = 3). Incubation time: 120 min for AMS, DAS, and AMDS, 180 min for DMTS, DADS, AMTS, and DATS. Dosage: 500 ng for AMS, DAS, AMDS, DMTS, and DADS; 25 µg for AMTS; 50 µg for DATS.

**Table 3 biomolecules-11-01874-t003:** The yield of each reactant after reaction with 400 µg of thymoquinone and each quantity of allyl mercaptan (AM).

Reaction pH	Time (min)	Quantity of AM (μg)	Content ± SD (ng/mL)		
			Reactant 1	Reactant 2	Reactant 3
1.2	10	0.5	n.d.	n.d.	n.d.
		5	0.14 ± 0.24	0.02 ± 0.04	n.d.
		50	4.05 ± 3.52	1.81 ± 0.88	0.08 ± 0.07
	30	0.5	n.d.	n.d.	n.d.
		5	0.16 ± 0.28	0.06 ± 0.10	n.d.
		50	8.12 ± 2.47	2.96 ± 0.85	0.22 ± 0.05
	60	0.5	n.d.	n.d.	n.d.
		5	0.28 ± 0.48	0.14 ± 0.24	0.02 ± 0.03
		50	10.08 ± 1.83	5.66 ± 0.22	0.68 ± 0.31
7.0	10	0.5	n.d.	0.26 ± 0.28	0.21 ± 0.20
		5	n.d.	5.58 ± 3.6	4.31 ± 1.23
		50	n.d.	422.69 ± 194.55	163.23 ± 38.93
	30	0.5	n.d.	1.42 ± 1.41	0.50 ± 0.55
		5	n.d.	18.42 ± 11.45	10.27 ± 0.77
		50	n.d.	605.32 ± 123.83	231.97 ± 49.12
	60	0.5	n.d.	2.13 ± 2.12	0.70 ± 0.63
		5	n.d.	28.38 ± 13.47	11.29 ± 0.32
		50	n.d.	732.26 ± 176.18	261.43 ± 36.45

n.d.: not detected. Each value represents the mean ± SD (*n* = 3).

## Data Availability

Data sharing is not applicable to this article.

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
