# Peer review of "Deodorant Activity of Black Cumin Seed Essential Oil against Garlic Organosulfur Compound"

_biomolecules, 2021, doi:10.3390/biom11121874_

Round 1
Reviewer 1 Report
Dear Editor,
After carefully reading the current manuscript I believe that it is very interesting, providing useful information concerning the major drawback of garlic human consumption which is bad breath. It seems that the authors possess excellent knowledge of garlic's special biochemistry and as well as chemistry, based on the fact most of the organosulfur compounds known for their beneficial bioactivity are not present in the intact garlic clove but on the contrary formed after enzymatic action and/or thermal conditions or treatments (in many cases different cooking aproach like boiling, frying, deep-frying, aging in hydroethanolic solutions, black garlic, etc. ). Also, experimentation, instrumentation, analytical methods and data presentation seems fair.
Nevertheless, I believe that some issues must be discussed before proceded for publication.
In detailed,
Line 16. Allyl mercaptan is not an ingredient of the intact garlic, on the contrary, it is been formed only after garlic tissue is been cut.
Line 31. Rephrase. 'Garlic is a species of Amaryllidaceae family".
Lines 167-168. On what basis do authors select these concentrations? Is it molar ratio between the reactants?
Lines 170-173. On what basis do authors select time periods that were different between GEO and various organosulfur compounds?
Line 182. Again, what was the basis for the reaction conditions 9mass versus volume ratio)?
Line 329. Why emphasize the description "...bulky compound"?
Line 330. Any suggestion about this observation?
Line 331. Change the word 'Traditionally.." with one more appropriate.
Line 337. At this point I am confused. In the end, is the deodorant activity of BCO caused by its hydrophobicity or by the action of its major constituent, thymoquinone?
Lines 382-383. Indeed, as it seems BCO may prevent from bud dreath consumers of garlic for healthy reasons, but on the other hand exactrly the prevened compounds by the action of thymoquinone are the ones responsible bor the beneficial effects on uman health! Please discuss.
Lines 382-383. Indeed, as it seems BCO may prevent garlic consumers' bad breath, but on the other hand, exactly the prevented, by the action of thymoquinone, sulfur compounds are the ones responsible for the beneficial effects of garlic on human health! Please discuss.
Reviewer 2 Report
The manuscript “Deodorant Activity of Black Cumin Seed Essential Oil against Garlic Organosulfur Compound” addresses the problem of halitosis due to garlic ingestion. Therefore, the deodorant activity of black cumin seed essential oil has been evaluated against volatile organosulfur compounds of garlic that cause bad breath after garlic ingestion.
The idea behind this work is interesting; but the scientific soundness seems scarce, as well as the experimental work. In particular, this work is more similar to a report instead of a scientific paper. An accurate discussion of the results obtained is missing, as well as a comparison with the previous literature. Indeed, the state of the art related to this problem is scarce and, therefore, the novelty of this study is not clear. The conclusions are poor and do not contain a critical analysis/overview of the findings. The English has to be improved and the manuscript is not organized according to the Journal template.
Round 2
Reviewer 1 Report
Dear Editor,
In this form, I believe that the present manuscript could proceed for publication.
Reviewer 2 Report
The authors improved the English; however, the scientific soundness remains scarce. More discussion of the results is required, as well as an accurate comparison with the literature.